# Hepatoprotective and Nephroprotective Effects of *Leea guineensis* Leaf Extract Against Paracetamol-Induced Toxicity: Combined Mouse Model-Integrated in Silico Evidence

**DOI:** 10.3390/ijms26136142

**Published:** 2025-06-26

**Authors:** Adedayo Titilayo Olukanni, Deborah Omotosho, Deborah Temitope Olalekan, Ernest Durugbo, Adeniyi Thompson Adewumi, Olumide David Olukanni, Salerwe Mosebi

**Affiliations:** 1Department of Biochemistry, Faculty of Basic Medical Sciences, Redeemer’s University, Ede 232101, Nigeria; olukannia@run.edu.ng (A.T.O.); feyidebie@gmail.com (D.O.); abiolat@run.edu.ng (D.T.O.); olukannio@run.edu.ng (O.D.O.); 2Department of Biological Sciences, Faculty of Natural Sciences, Redeemer’s University, Ede 232101, Nigeria; durugboe@run.edu.ng; 3Department of Life and Consumer Sciences, College of Agriculture and Environmental Sciences, Florida Campus, Roodepoort 1710, South Africa; adewuat@unisa.ac.za

**Keywords:** *Leea guineensis*, paracetamol, hepatotoxicity, nephrotoxicity, oxidative stress

## Abstract

Acetaminophen, or paracetamol (PCM), is a common painkiller used to treat aches, pain, and fever. Nevertheless, PCM has been reported to be hepatotoxic and nephrotoxic in humans. Thus, there is a need to identify how this side effect can be treated. Previous studies have shown that Leea species possess antioxidative, anthelmintic, anti-cytotoxic, hepatoprotective, and nephroprotective properties. However, the role of *Leea guineensis* (LG) in modulating PCM-induced hepatotoxicity or nephrotoxicity remains unknown. Herein, we investigate the possibility of *Leea guineensis* leaf extract (LGE) to ameliorate PCM toxic effects, evaluate hepatic and renal function, oxidative stress markers, and safety, and perform molecular docking to predict affinities of *Leea guineensis* extract compounds for their targets compared to PCM. An in vivo rat model was used for Leea guineensis extract or silymarin (SLM, standard drug) at various concentrations, and it was co-administered with PCM. We observed that *Leea guineensis* extract is rich in phytochemical constituents, and its treatment in rats did not significantly affect body weight. Our data showed that PCM increased bilirubin, creatinine, uric acid, Alanine aminotransferase (ALT), and cholesterol levels but decreased Aspartate aminotransferase (AST) in plasma. Moreover, it increased lipid peroxidation (MDA) levels in the liver and kidneys, while the total protein was elevated in the latter. Interestingly, *Leea guineensis* extract and SLM abrogated the elevated parameters due to PCM toxicity. Importantly, histopathological examination showed that *Leea guineensis* extract demonstrated the potential to ameliorate hepatic and renal lesions caused by PCM intoxication, thus demonstrating its safety. Furthermore, comparative molecular binding affinities of the study ligands binding the target corroborate the experimental findings. Our study shows that *L. guineensis* leaf extract, through its rich phytochemicals, can protect the liver and kidneys against the toxic effects of paracetamol in a dose-dependent manner.

## 1. Introduction

The liver is an organ that detoxifies, synthesizes, and produces biomolecules involved in digestion and growth [1]. Hepatotoxicity, however, results in liver inflammation and damage to the hepatocyte, causing the leaching of biomolecules, such as bilirubin, aspartate aminotransferase (AST), alanine aminotransferase (ALT), and alkaline phosphatase (ALP), produced or stored in the liver [2]. If the toxicity is left untreated, it may cause permanent liver scarring or cirrhosis, which precedes liver failure, and death is inevitable [3]. Another vital organ in detoxification is the kidney, which controls body fluid volume, osmolality, acid–base balance, electrolyte concentrations, and toxin elimination. However, when toxins, hazardous chemicals, or medications are administered at high concentrations, damages often occur in the nephrons. Nephrotoxicity is characterized by a rapid rise in serum creatinine, a decrease in glomerular filtration rate, and moderate arterial hypertension [4]. In renal biopsies, tubular, arterial, and glomerular alterations and interstitial fibrosis can be seen. Acute renal failure, persistent interstitial nephritis, and nephritic syndrome are three clinical syndromes associated with drug-induced nephropathy. Because there are no signs, the damage to these organs can take years to manifest [5]. Medications can harm the liver and the kidneys in various ways, from a moderate aberrant function like elevated serum aminotransferase and creatinine levels to severe organ destruction like hepatocellular necrosis, nephrosis, and nephritis [6,7]. Paracetamol is a commonly used analgesic and antipyretic medication; it is an over-the-counter medicine that can be widely marketed without supervision, and it can cause liver and kidney damage in excessive dosages [6,8]. Despite advances in contemporary medicine, few synthetic medications are available to treat liver disease. However, several plants treat liver problems therapeutically [9]. Silymarin, an extract from milk thistle, has been documented to have both hepatoprotective and nephroprotective activities [7], and it is currently in use as a standard drug for treating liver inflammation or damage. Therefore, more plant extracts must be investigated for their ability to protect the liver and kidneys. Leea species are a family of shrubs native to Africa and common in Australia, New Guinea, and South and Southeast Asia; about 70 species of this Vitaceae family of tiny trees or shrubs have been documented. Several biological studies have described the antibacterial, antioxidative, anthelmintic, cytotoxic, hepatoprotective, and nephroprotective properties of the Leea species [10]. Leea species have been used in the orthodox treatment of enlarged spleens in children, pregnancy detection, purgative, toothache, gonorrhea, general weakness, skin lesions, skin rash, ulcers, herpes, and boils. In oral treatments, they have also been used as painkillers, for epileptic fits, as diuretics, and in the treatment of convulsions, spasms, diarrhea, and dysentery. The antifungal and antibacterial properties of Leea species have also been documented [11]; there are, however, scant reports on the bioactivities of *Leea guineensis* vis-à-vis acetaminophen-induced organ damage. Acetaminophen-induced liver injury has been associated with glutamate-cysteine ligase activity, the enzyme that catalyzes the rate-limiting and first step in glutathione synthesis [12]. For the enzyme to be effective, it must be modified at the transcriptional level; transcriptional modification of glutamate-cysteine ligase GLC is essential for its overexpression during toxic situations, the center of which is the Nrf2. Nrf2 exists in the cytoplasm as a complex with Keap1 homodimer, and it is released before its movement into the nucleus, where it stimulates antioxidant genes, with GCL genes being prominent. Two models have been proposed to release Nrf2 from the Keap1 homodimer: the CUl3 dissociation and the ‘hinge and latch’ models. In the dissociation model, ROS oxidized the cysteine residue of Kaep1, releasing Nrf2. In contrast, hinge and latch models proposed that oxidation enables the irreversible binding of the Keap1 DGR domain to Nrf2, thereby preventing the binding of newly synthesized Nrf2 [13]. The new Nrf2 then enters the nucleus, modulating the transcription of antioxidant genes. Publications have shown derivatives of dithiolethiones that target Nrf2 activation [14]. We thus proposed that phytochemicals with binding similarities to the dithiolethiones would activate Nrf2 release from Keap1 or prevent the binding of new Nrf2, therefore exhibiting protective activities against toxicity. This study is thus designed to investigate the ability of *Leea guineensis* leaf extract to protect the liver and the kidneys against paracetamol-induced organ toxicity and to investigate the interaction of the prominent constituents of the plant’s leaves in glutathione synthesis using in silico molecular docking. Molecular docking is a technique that finds applications in various fields, especially in drug design. For this study, the technique is employed to investigate the mechanistic knowledge and insights into how *Leea guineensis* bioactive compounds may reduce oxidative stress at the molecular level by identifying potential molecular targets of paracetamol toxicity and how *Leea guineensis* phytochemicals might interact with them. This improves clinical relevance, even though in vivo methods are essential for comprehending systemic toxicity and pharmacodynamics. To evaluate therapeutic promise, expedite drug discovery, and support customized methods, it aids in the understanding of binding affinities, interactions, and inhibitory potential. The integration of both methods provides a comprehensive and functionally relevant assessment of liver injury in the context of paracetamol-induced toxicity.

An animal model was used to simulate the drug-related organ toxicity because it can accurately provide the disease or condition under study and enable the observation of treatment effects, disease progression, or biological mechanisms. The model also affords quantitative and qualitative data analysis of the set objectives while replicating the biological or physiological processes since the rats have genetic, anatomical, or metabolic similarities to humans, for which conclusions are to be drawn.

## 2. Results

### 2.1. Phytochemical Screening of Leea guineensis Extract

Qualitative phytochemical screening of *Leea guineensis* ethanolic leaf extract revealed the presence of alkaloids, flavonoids, saponins, steroids, and tannins (Table 1). Carbohydrates, proteins, and glycosides were also present in the *L. guineensis* extract. The extract showed high levels of tannins, flavonoids, and glycosides.

### 2.2. Effect of Treatment on Body and Organ Weight

The animals’ body and organ weights are shown in Table 2. There were no significant differences in the mean group weights of the experimental animals. The final mean body weight of the PCM + 900 mg LGE (177.80 ± 17.41) was not significantly different from that of the healthy animals (193.40 ± 17.74). There are also significant differences in the relative weight of animals with standard drugs and the various doses of tested LGE compared with the PCM toxic group. Toxicity was induced in the animals with 3000 mg/kg PCM, resulting in a non-significant decrease in liver and kidney weights (2.56 and 6.34%, respectively). A significant increase was, however, noticed in the SLM (Standard drug) groups’ liver weights against significant decreases experienced in the 300 mg and 600 mg LGE treatment groups. A similar but not significant reduction in kidney mean weight was encountered with the SLM drug. An initial decrease in weight was experienced with the LGE 300 mg group, followed by a dose-dependent increase in kidney weight for the LGE 600 and LGE 900 groups.

### 2.3. Hepatotoxicity of PCM and Effect of Leea guineensis Extract

We measured plasma AST, ALT, bilirubin, and cholesterol levels to explore whether Leea guineensis extract would reduce PCM-induced hepatotoxicity (Figure 1). The data showed that PCM increased ALT, bilirubin, and cholesterol levels while AST levels decreased compared to the control group. Interestingly, both the standard drug (silymarin) and the test substance (*Leea guineensis* extract) were able to reduce the PCM-induced elevated ALT level in a dose-dependent manner (Figure 1A). Surprisingly, the different doses of *Leea guineensis* extract showed a more pronounced effect in reducing PCM-induced ALT levels than silymarin. On the contrary, PCM reduced AST levels compared to the control group, while silymarin and *Leea guineensis* extract increased AST levels (Figure 1B). Treatment with PCM significantly increased bilirubin levels, while silymarin had no significant effect; 600 mg/Kg and 900 mg/kg LGE reduced PCM-induced bilirubin levels (Figure 1C). A similar significant increase in plasma cholesterol levels was experienced with rats exposed to PCM (Figure 1D). Still, SLM and *Leea guineensis* extract (at all doses) reduced the PCM-induced high cholesterol level.

### 2.4. Effect of Leea guineensis Extract on PCM-Induced Nephrotoxicity

We then asked whether *Leea guineensis* extract would control PCM-induced nephrotoxicity by evaluating the creatinine, urea, and uric acid plasma levels. The result of nephrotoxicity induction shows that PCM administration increased these parameters in the plasma (Figure 2); however, only creatinine and uric acid were significantly increased, while urea showed a trend toward increased levels. We observed that both SLM and *Leea guineensis* extract could suppress the elevated creatinine levels caused by PCM (Figure 2A). The *Leea guineensis* extract appears to have a better ability to reduce creatine-induced PCM than silymarin. The different doses of *Leea guineensis* extract also caused a significant reduction in plasma urea levels; meanwhile, SLM caused a further increase in urea levels compared to PCM treatment (Figure 2B). Moreover, the elevated uric acid caused by PCM administration was suppressed significantly by both silymarin and except in the group that received 600 mg/kg *Leea guineensis* extract.

### 2.5. Effect of Leea guineensis Extract on PCM-Induced Oxidative Stress

Hepatotoxicity and nephrotoxicity were ascertained in the liver and kidney by measuring the alteration of total protein (TP), lipid peroxidation (MDA), catalase (CAT), superoxide dismutase (SOD), glutathione peroxidase (GPx), glutathione reductase (GSH), and glutathione -s-transferase (GST) (Table 3). In the liver, treatment with paracetamol caused a significant (*p* < 0.05) increase in MDA, CAT, and GST. At the same time, the SOD and the total protein decreased significantly (*p* < 0.05) compared to the control group. Co-administration of PCM and SLM shows that the standard drug affected the impact of PCM on MDA (30.40% decrease), CAT (19.20% decrease), SOD (370.00% increase), GPx (8.67% decrease), GSH (7.70% increase), and GST (236.00% increase). Whereas, the tested drug (LGE) at 300 mg/kg significantly reversed the activities of PCM on MDA, SOD, and GPx. LGE administration at 900 mg/kg significantly alleviated (*p* < 0.05) the effect of PCM on the total protein (139.33% increase), MDA (46.49% decrease), CAT (58.26% decrease), SOD (60.00% increase), and GPx (36.00% decrease). In all these parameters, only the SOD and GSH were returned to normal levels by the SLM, while the *Leea guineensis* extract returned the SOD (LGE300 and LGE 600) and GST (LGE 600 only) to normal levels, as seen in the control animals. Similarly, LGE 900 mg/kg returned TP, GPx, and GSH to normalcy (*p* < 0.05). The results of the oxidative stress induction of paracetamol in the kidney show a similar trend to that of the liver; the PCM group shows a significant (*p* < 0.05) increase in total protein and MDA, while the CAT and SOD levels were reduced (*p* < 0.05) compared to the control group (Table 3). Exposure of the animals to SLM post-PCM treatment shows that the silymarin reduced the impact of PCM on TP (71.2% decrease), MDA (37.6% decrease), CAT (71.4% increase), and GST (62.5% increase). Meanwhile, LGE at 600 mg/kg significantly reversed the activities of PCM on MDA, CAT, and GPx. LGE administration at 900 mg/kg significantly altered the effect of PCM on TP (48.8% decrease), MDA (78.5% decrease), CAT (80% increase), GPx (45.65% decrease), and GST (50% increase). While the SLM reversed the SOD, GSH, and GST to the normal range, both 300 mg/kg and 600 mg/kg of *Leea guineensis* extract reversed SOD and GST, and the 900 mg/kg concentration reversed SOD, GSH, GST, and GPx to the control group range. Although the doses of *Leea guineensis* extract could not bring the kidney levels of TP and MDA to normal values, there were apparent dose-dependent reductions in the PCM-LGE groups compared to the PCM groups.

### 2.6. Histopathological Evaluation

The histology of rat livers from the six groups was observed under a light microscope with 100× and 400× magnification. The hepatocytes, sinusoids, and portal triad (hepatic vein, hepatic artery, and bile duct) are visible across the various groups. The thick red arrow denotes areas with observed significant alterations. The rat liver cells in healthy controls appear similar to those in *Leea guineensis* extract-treated groups. Rat liver cells in PCM showed damaged cells with red coloration (Figure 3B). The healthy control group shows normal hepatic cells, while inflammation and necrosis show intoxication due to paracetamol induction. However, treatment with extracts of *Leea guineensis* reversed the hepatic lesions caused by paracetamol intoxication to a large extent.

Histological profiles of kidney cells in 6 rat cohorts are presented in Figure 4. Normal control rats (Figure 4A) showed normal glomerular and tubular histology in the paracetamol-induced nephrotoxicity model. In contrast, the toxic control showed distorted tubular shape, cellular infiltration of the tubules, glomerular and blood vessel congestion, and inflammatory cells in kidney sections (Figure 4B). The treatment with silymarin and LGE of different doses reduced such changes in kidney histology Figure 4C–F. The renal cortex, renal tubules, glomeruli, mesangial cells, renal parenchymal cells, and glomerular capsules are all visible across the various groups. The thick red arrow denotes areas with observed significant alterations.

### 2.7. Molecular Docking

Comparative molecular docking studies were carried out to gain further insights into the activities of the ligands (*Leea guineensis* extract (Figure 5) and PCM) binding to their targets [15]. Generally, the molecular docking technique provides binding affinity predictions between a ligand and a target, which in this study case, was extrapolated to the creatinine and urea levels in this manuscript. These proteins were chosen because they protect cells against drug toxicity and oxidative stress. Keap1 regulates antioxidant and detoxification pathways by binding to Nrf2, a transcription factor. The induction of HO-1 protects against drug-induced toxicity and oxidative stress. NQO1 is a flavoprotein that reduces quinones and other electrophiles, preventing oxidative stress and toxicity, and its activation protects against drug-induced toxicity and carcinogenesis. GCLM is a subunit of glutamate-cysteine ligase, the rate-limiting enzyme in glutathione synthesis, and increased GCLM expression enhances glutathione levels, protecting against drug-induced oxidative stress and toxicity. Modulation of Cullin-3 expression influences drug-induced toxicity and Nrf2-mediated defense mechanisms.

Table 4 provides the docking binding affinity/energy (BA/BE) heatmap in kcal/mol. The heatmap table used a green–red highlight to indicate the highest negative energy values. In contrast, the lowest negative energy values are highlighted in red. In comparison with the reference ligand, PCM (highest BA = −6.2 across the targets) and the *Leea guineensis* extract compounds revealed relatively higher binding affinity values with all the target proteins, except 1,2,3-benzene triol, Pyrazole-5-carboxylic acid, and n-hexadecanoic acid, indicating more robust binding and stability [16]. However, the result showed that beta-sitosterol has the highest binding affinity (–9.7 kcal/mol) with KEAP 1 compared to other ligands with all the proteins, except heme oxygenase 1, as indicated by its high BE for each protein. Squalene and Benzo(h)quinolinine showed the highest BA values of –8.3 and –8.2 kcal/mol, respectively. Moreover, Curan 17-oic acid and vitamin E also showed interesting binding affinities for all the proteins. Lastly, the results imply poor ligand binding with cullin 3, except for beta-sitosterol, with the lowest BA (–7.0 kcal/mol). In addition, Vitamin E, beta-sitosterol, and curan 17-oic acid gave BA values above −9.0 kcal/mol binding to quinine oxidoreductase 1 (NQO1). Thus, this indicates that the protein at this conformation (PDB ID 3JSX) is preferably a potential target among the study proteins for Vitamin E, beta-sitosterol, and curan 17-oic acid. More importantly, the significantly higher BA values of squalene, Vitamin E, beta-sitosterol, betatocopherol, Thieno (2,3-C) furan-3-carbonitrile, and curan 17-oic acid compared to PCM agree with the experimental findings that LGE reduces PCM levels in this study and especially indicate a promising design of anti-inflammatory treatment. The docking results confirmed that the binding affinities (BAs) of the compounds available in the extracts were higher than the PCM’s BA, except 1,2,3-benzenetriol and n-hexadecanoic acid. Therefore, the use of the *Leea guineensis* extracts as antioxidative, anthelmintic, cytotoxic, hepatoprotective, and nephroprotective against PCM has potential benefits because the compounds bind stronger to the targets than PCM.

To better understand the binding affinity of the ligands, we obtained the interaction networks in Figure 6A–C of beta-sitosterol with KEAP 1, Heme Oxygenase 1, NQO1, and GCLM and squalene interaction forces with Heme Oxygenase 1 in Figure 6D,E. Corroboratively, the interactions illustrate the mechanisms of binding or prospective inhibitory actions of five binding ligands to their respective targets. Beta-sitosterol formed alkyl interactions with KEAP 1 via a highly positively charged arginine (ARG) residue and a strong hydrophobic alanine (ALA). The different colors provide a quick understanding of the values as higher negative values (more favorable docking scores) are depicted in shades of green while lower negative values are depicted in red. 

Moreover, squalene formed strong alkyl interactions with Heme-Oxygenase 1 via highly hydrophobic residues, alanine (ALA), valine (VAL), phenylalanine (PHE), and leucine (LEU) and additional pi-alkyl interactions with methionine (MET). Interestingly, only the interactions between the beta-sitosterol and NAD(P)H Quinone formed additional types of strong interactions, including conventional H-H bond, C-H bond, pi-pi stacked, and pi-pi T-shaped, which justify why the energy and stability are favored in this complex. Penultimately, we observed various strong alkyl and pi-alkyl interactions formed between beta-sitosterol and Glutamase-Cysteine ligase via the hydrophobic residues lysine (LYS), tryptophan (TRP), methionine (MET), and electrically charged histidine side chains. Lastly, the weak interactions between beta-sitosterol and cullin 3 showed the reason for the significantly low binding affinity value (–7.0 kcal/mol) compared to the BA of beta-sitosterol with other target proteins.

## 3. Discussion

The present study considered the protective possibility of *Leea guineensis* leaf extract, a traditional medicinal shrub, against paracetamol-induced liver and kidney damage. To investigate the possibility of the protection of plant extracts, hepatotoxicity and nephrotoxicity were induced in rats after 12 days of administration of SLM and LGE using 3000 mg/kg body weight of PCM. The effect of *Leea guineensis* extract was tested using the activities of AST, ALP, and bilirubin in rat plasma, which was induced by paracetamol. Hepatotoxic activities of paracetamol at high doses are well documented [17,18]. About 25% of administered paracetamol is bound to plasma proteins, while the rest enters the liver, where liver microsome enzymes partially metabolize it. In the liver, paracetamol can be conjugated with glucuronic acid and converted by cytochrome P450 to N-acetyl-p-benzoquinonylmin, a highly reactive metabolite. This metabolite is detoxified quickly by conjugating with the cell glutathione; paracetamol also binds to mitochondrial proteins, damaging ATP synthesis and releasing reactive oxygen species [19]. In this way, the levels of plasma proteins, liver glutathione, and enzymes are expected to be adversely affected by a high paracetamol dosage. Furthermore, the reactive oxygen species generated confer oxidative stress on the organ, causing hepatotoxicity [20].

The presence of tannins, flavonoids, and glycosides in the phytochemical screening of *Leea guineensis* extract suggests it would be a good candidate for controlling hepatotoxicity and nephrotoxicity. The antioxidant nature of flavonoids is well documented [21,22]. Zhang et al. [23] have reported that tannic acid’s hepatoprotective mechanisms could be associated with its anti-oxidation, anti-inflammation, and anti-apoptosis activities. Certain glycosides have been reported to be nephroprotective but have components that could aggravate hepatotoxicity [24]. The extracts in this study exhibited both hepatoprotective and nephroprotective activities, which may be linked to the whole plant leaves with diverse constituents against the single molecules used in the abovementioned studies. The different components of the plant extract may exhibit a synergetic effect, thus masking the toxic impact of single molecules. The mean animals’ equal weight implies that SLM and *Leea guineensis* extract have no adverse effect on the rats’ appetite and eating pattern since the drugs were administered for 12 days. The same observation with the PCM groups may result from the short time between administration and study endpoint.

The increase in ALT, bilirubin, and cholesterol in the plasma and a discrete decrease in AST levels are evidence of the pathological condition induced by PCM. AST, ALP, and bilirubin, which are biomolecules found primarily in liver cells and trace amounts in the blood; however, if the liver cell membrane is broken, these molecules leak into the bloodstream. This situation results in elevated blood levels of the enzymes/molecules, thus indicating a rupture of the liver cell membrane. ALT is more specific to the liver, while AST is found in multiple tissues (e.g., liver, heart, muscle, kidney, brain). A rise in ALT with normal or decreased AST may indicate localized or mild liver injury, where hepatocytes are affected without extensive necrosis, or this may be because ALT is released before AST due to its greater liver specificity and longer half-life [2]. The ability of the standard drug (silymarin) and *Leea guineensis* extract to reverse the plasma parameters to a range like that of healthy rats shows these drugs’ hepatoprotective and nephroprotective activities. Although the therapeutic activities of the Leea genus have been reported, only a few exist on *Leea guineensis*. Most of these accounted for their traditional applications in treating skin rash, ulcer, diarrhea, paralysis, toothache, rheumatism, skin ulcer, vertigo, epileptic fits, and paralysis [10,25,26]. *M. cecropioides* hydromethanolic leaf has been reported to reduce the increased AST, ALT, and bilirubin levels induced by CCl_4_ intoxication [27]. The plasma urea, creatinine, and uric acid parameters were assessed as kidney function. A rise in blood creatinine is a sign that renal function is deteriorating. Paracetamol-intoxicated rats had higher plasma urea, uric acid, and creatinine concentrations than the control group. Compared to the control group, plasma urea, uric acid, and creatinine concentrations were increased significantly (*p* > 0.005). These findings are virtually identical to those of [28], who found that continuous injection of Cyclosporine A for 21 days significantly decreased renal function. When the ethanol extract of *Leea guineensis* was given to paracetamol-treated rats, the urea, uric acid, and creatinine levels were lower than in the paracetamol group.

Cells contain an intrinsic defensive mechanism, including antioxidant defense machinery, that maintains cellular oxidative balance by removing endogenously generated free radicals or foreign chemical exposure during normal physiological activities. The enzymatic antioxidants SOD, CAT, and GPX act together to protect cells from oxidative stress. The increase in the liver’s MDA, CAT, and GST and the decreased SOD and total protein are evidence of the hepatotoxicity of PCM, probably induced by oxidative stress, while the improvement experienced on co-administration of PCM and SLM shows that the standard drug can reduce the toxic effect of paracetamol on the liver. The tested substance also ameliorated the hepatotoxicity effect of PCM, with the overall best result when a *Leea guineensis* extract dosage of 900 mg/kg was administered. The therapeutic potential of LGE at 900 mg/kg appeared to be better than that of the standard drug since the total protein, lipid peroxidation, and catalase levels altered by PCM toxicity were reverted to normal values, compared to glutathione reductase, which was reverted by the standard drug. SOD dismutase is the superoxide anion to hydrogen peroxide. In contrast, glutathione peroxidase detoxifies hydrogen peroxide to oxygen and water because of the decrease in SOD and GPX activity in rats given paracetamol, as superoxide radicals and hydrogen peroxide could build up in the hepato-renal system [29].

Paracetamol also induced nephrotoxicity in this study, while co-administration with the SLM drug reversed four parameters: TP, MDA, CAT, and GST. *Leea guineensis* extract at 900 g/kg body weight reversed the PCM effect of five parameters, adding GPX to the list. This reversal of alteration of the antioxidant markers suggested the SLM drug’s and *Leea guineensis* extract’s antioxidant properties. GST has been implicated in phase II detoxifications, hormone synthesis, tyrosine degradation, stress signaling, oxidation–reduction reactions, and the posttranslational glutathionylation of proteins [30]. Increased catalysis of the conjugation reaction between PCM and GSH as part of cellular defense against free radicals could explain the increased GST activity and consequent depletion of GSH after PCM exposure [31]. *Leea guineensis* had a similar therapeutic effect on hepatic and renal GST and GSH levels in PCM-intoxicated rats. Furthermore, the anti-lipid peroxidative capability of *Leea guineensis* was demonstrated by its considerable reduction in the PCM-mediated increase in MDA [32].

Molecular docking is one tool that helps predict the binding affinities of ligands to specific proteins, thereby providing more insight into the biological activities of these ligands [33]. Molecular docking was employed in order to further understand the mechanisms by which the leaf extract of *Leea guineensis* provided significant hepatoprotective and nephroprotective activities against paracetamol-induced toxicity in rats. In order to obtain the best result, the ligands employed were docked at the active site of the enzymes, where the maximum catalytic activity occurs, which also helps to accurately predict the ligand–enzyme interactions. As observed from the results obtained from the docking, beta-sitosterol had a high binding affinity with all the proteins employed for the docking. The proteins studied were those that are known to play important roles in protecting against the effects of toxicity through the stimulation of enzymes and factors with antioxidant activities. The high binding affinity of three prominent ligands in the extract beta-sitosterol, curan 17 oic acid, and vitamin E with all the proteins helped to elucidate the mechanism by which the leaf extract of *Leea guineensis* provided the hepatoprotection and nephroprotection observed in vivo as it implies that the aforementioned ligands which are part of the constituents of the leaf extract were able to activate the proteins studied, which resulted in their protective activities against toxicity. The high negative energy of beta-sitosterol (−9.7) and curan 17-oic acid (8.5) with KEAP1 is worth noting. Beta-sitosterol showed good to excellent binding affinity to all the proteins tested, while squalene gave the highest binding energy with heme oxygenase 1. Paracetamol, which was used as a reference control, showed relatively weak interactions with the proteins as indicated by its relatively low negative binding energy values, which could also be a likely mechanism by which it induces oxidative stress in vivo.

KEAP1 is a homodimeric protein belonging to the BTB (Broad complex, Tramtrack, Bric-á-brac) Kelch family of proteins named Kelch-like 1 to 42 (KLHL1–42). It functions as a substrate adaptor for a Cullin 3 (CUL3)-based E3 ubiquitin ligase that targets NRF2 for ubiquitination and proteasomal degradation in the absence of oxidative stress [34]. Nrf2 is a key transcription factor that activates cytoprotective genes, including those encoding antioxidant enzymes and enzymes involved in detoxification. Keap1 is an adaptor protein of Cullin3-based E3 ligase, which regulates Nrf2 activity when there is oxidative or xenobiotic stress, such as electrophiles. ROS can change the cysteine residues of Keap1 to inactivate the ubiquitin E3 ligase activity of Keap1 so that Nrf2 escapes from the Keap1-mediated repression, migrates into the nucleus, and activates the expression of its target genes [35]. Activated Nrf2 then moves into the nucleus, is heterologous to the small musculoaponeurotic fibrosarcoma (sMAF) protein, and binds to ARE (antioxidant response element) [36]. ARE is in the promoter region of multiple genes that encode phase II detoxifying enzymes and antioxidant proteins [37]. ARE is crucial for the transcriptional activation of antioxidant genes such as NQO1, GSTs, and glutamate-cysteine ligase. In this light, the high affinity of beta-sitosterol and curan-17-oic acid as electrophiles for KEAP 1 might have allowed the activation of Nrf2, which subsequently activated NQ01 and GCLM (glutamate-cysteine ligase modifier), thereby offering significant protection against paracetamol-induced toxicity as observed from the results obtained in this study. Furthermore, GCLM is known to reduce the K_M_ (Michaelis constant) value for the substrate glutamate and raise the K_i_ (inhibitor constant) value for GSH, thereby optimizing the conditions for the synthesis of glutathione, a known antioxidant [38]. The high binding affinity of beta-sitosterol for GCLM might have also induced the production of glutathione, an intracellular defense against paracetamol-induced oxidative stress.

NQO1 is an enzyme reported in various earlier studies to be involved in a unique antioxidant role, mainly by endogenous quinone reduction, which defends cellular membranes against lipid peroxidation. It protects the cell from unwanted oxidation by maintaining the reduced form of endogenous antioxidants. In vitro studies have shown that alpha-tocopherol (vitamin E) is a significant substrate for NQO1 [39], corroborating our results, as vitamin E and beta-tocopherol had relatively high binding energy values for the enzyme. Furthermore, NQO1 is directly able to scavenge superoxide; hence, the activation of this enzyme by other ligands such as beta-sitosterol, beta tocopherol, and curan 17-oic acid, which also gives high binding affinity, could provide extra protection against toxicity-induced oxidative stress.

Heme oxygenase 1 (HO-1), an inducible enzyme responsible for heme degradation, has been reported to possess antioxidant and anti-inflammatory properties [40]. Squalene, which gave the highest binding free energy with HO-1, is a potent antioxidant attributed to its many double bonds present in its structure [41]. The presence of squalene in the extract indirectly contributed to the antioxidant activity observed due to the stimulation of heme oxygenase by the binding of squalene to the enzyme.

Cullins are a family of proteins that confer substrate specificity to multimeric complexes of E3 ligases acting as scaffold proteins [42]. An array of studies has emphasized the importance of Cullin-3 (Cul3) in the regulation of Nrf2 activity since it has been found that this transcription factor is present in Cul3 complexes in vivo, which constitutively leads to Nrf2 degradation via the UPS (ubiquitin-proteasome system) [43]. Cul3 has been implicated in regulating the activity of Nrf2 during stress responses. Cul3 has been reported to be more specific for the regulation of certain substrates, as well as the reduction of toxicity because it does not inhibit bulk proteasomal degradation [42]. The activation of cullin three by beta-sitosterol, which gave the highest binding free energy for cullin 3, could have suppressed paracetamol-induced toxicity, thereby providing resistance and protection for the tissues directly affected.

Furthermore, paracetamol administration substantially impacted the histological damage to the livers and kidneys of the rats in the paracetamol group. Once again, the ability of different doses of *Leea guineensis* extract to protect against paracetamol intoxication was evident in the histological images. The images are better in the kidney; inflamed cells were found in the organs of the unprotected paracetamol-induced rat, indicating interstitial inflammation. Bencheikh et al. [4] reported fewer glomerular cells, interstitial fibrosis, and vascular congestion, resulting in tubular epithelial atrophy during drug-induced toxicity. The kidneys of the rats treated with *Leea guineensis* extract in conjunction with paracetamol showed no morphological alterations. Overall, the hepatoprotective and nephroprotective actions of *Leea guineensis* ethanol extracts compared favorably with those of the standard drug (silymarin) used in this study. It is believed that the results from this study would apply to humans; we expect to have similar responses in human beings since the pathophysiology of rats is identical to that of humans.

## 4. Materials and Methods

### 4.1. Chemicals and Assay Kits

Alanine aminotransferase kit, aspartate aminotransferase, bilirubin, urea, creatinine, uric acid, and cholesterol were purchased from Randox Laboratories, Antrim, UK, but were purchased from Nigerian distributors, Lagos Island, Lagos. Some of their Catalogue code includes AS 101, AL 100, UR 1068, and BR 411. Tricarboxylic acid and Griess reagent (Sigma Aldrich, St Louis, USA), reduced glutathione and Ellman’s reagent (CDH, Daryagani Delhi, India) Bovine serum albumin and epinephrine (Janssen, Leics, UK), were purchased by the school’s procuring officer for the laboratory. Paracetamol (500 mg) (Emzor, Lagos, Nigeria) and silymarin (70 mg) (Orion, Gujarat, India) were purchased from a pharmacy in Owode-Ede, and KCl 1.15%, phosphate buffer 0.1 M, NaOH 0.2 M, 0.05 M carbonate buffer, TCA 10%, TBA 0.75% and 0.1 M HCl (Molychem, Mumbai, India) were of analytical grade.

### 4.2. Collection and Identification of Plant Samples

Fresh *Leea guineensis* leaves were obtained from Redeemer’s University campus, Ede Osun State, identified by a botanist at the Department of Biological Sciences Redeemer’s, University Ede, and confirmed at the botany unit at The University of Lagos, where a sample was deposited and a voucher number, LUH 8195, was provided. The leaves were air-dried and macerated in a blender.

### 4.3. Plant Extract Preparation

*Leea guineensis* (500 g) was soaked in 1500 mL absolute ethanol and put on the shaker for 72 h. The supernatant was filtered with Whatman filter paper. No. 4 and concentrated with a rotary evaporator, then dried in a vacuum oven at 40 °C. The dried extract weighing 55 g gave an extraction rate of 11% using the formula below, as follows:(1)$$Yield\%=100\times \frac{weight\;of\;dried\;extract}{weight\;of\;dried\;plant}$$

The extract was then stored at 4 °C for further analysis.

### 4.4. Phytochemical Analysis

The qualitative phytochemical evaluation was conducted on the extract to determine the presence of alkaloids, flavonoids, glycosides, phenols, phlorotannins, saponins, tannins, carbohydrates, and proteins according to standard methods [44,45]. Color, foam, and precipitate formation were used to indicate positive responses in tandem with the respective test.

### 4.5. Animals

Healthy male Sprague–Dawley rats were purchased from Animal Farm, Ibadan, housed in plastic cages, and fed with commercial rat chow obtained from Ladokun Feeds, Ibadan, Nigeria. After two weeks of acclimatization, 30 rats weighing approximately 170–200 g were grouped into six groups, and rat chow and water were provided ad libitum under a 12 h light/dark cycle at room temperature at Redeemer’s University animal house, following the Redeemer’s University Committee on Ethics for Scientific Research guidelines on laboratory animal handling. In addition, the lead author has taken and passed the Collaborative Legislation Training Initiative for the West African Bioethics Training Program on Animal Biosafety. The Redeemer’s University Committee on Ethics for Scientific Research approved the study and assigned RUN/BCH/17/17013 as the authorized number.

### 4.6. Experimental Design

The rats were distributed into six groups, with five animals per cage;Group A: CTR, received normal saline only;Group B: PCM, received a single dose of 3000 mg/kg paracetamol;Group C: SLM, received a single dose of 3000 mg/kg paracetamol plus 200 mg/kg silymarin;Group D: LGE300, received a single dose of 3000 mg/kg paracetamol plus LGE at 300 mg/kg;Group E: LGE600, received a single dose of 3000 mg/kg paracetamol plus LGE at 600 mg/kg. Group F: LGE900, received a single dose of 3000 mg/kg paracetamol plus LGE at 900 mg/kg. All treatments administered to animal groups were performed orally using an oral cannula, and the animals were sacrificed on the 14th day, 2 days after the administration of PCM as described by Sinaga et al. [46].

Note: CTR: control group, PCM: Paracetamol, SLM: Silymarin, LGE: Leea guineensis extract

### 4.7. Drug Preparation and Toxicity Induction

Both silymarin and paracetamol powder were dissolved in normal saline. While silymarin was administered at 200 mg/kg of rat body weight, paracetamol was administered at 3000 mg/kg. The induction of hepatotoxicity was performed using the Sinaga et al. [46] method. Paracetamol suspensions were administered orally to groups B–F on day 12. Hepatotoxicity was determined by measuring the following biochemical markers: elevated serum levels of ALT (Alanine aminotransferase), AST (Aspartate aminotransferase), bilirubin, oxidative stress markers such as reduced glutathione (GSH) levels in liver tissue, increased malondialdehyde (MDA), which is a marker of lipid peroxidation, and altered antioxidant enzymes, namely superoxide dismutase (SOD), catalase (CAT), and glutathione peroxidase (GPx).

### 4.8. Animal Handling and Tissue Harvesting

The rats were sacrificed by cervical dislocation, and blood was collected from the pumping heart using a 5 mL syringe; the blood sample was kept inside a labelled EDTA sample bottle and centrifuged at 4000 rpm for 5 min to obtain plasma. The liver and kidney were harvested from the rats, blotted of bloodstains, rinsed in 1.15% KCl, and then homogenized in 0.1 M phosphate buffer (pH 7.4). The homogenates were centrifuged at 12,500× *g* for 15 min at 4 °C, and the supernatants were used for enzyme assays.

### 4.9. Measurement of Biochemical Parameters

According to the manufacturer’s instructions, nephrotoxicity was determined by measuring creatinine and urea levels in plasma using the Randox Kits (UK). Total bilirubin, albumin, aspartate aminotransferase (AST), and alanine aminotransferase (ALT) enzymes were also determined in the plasma to evaluate hepatotoxicity using kits from Randox (UK). The total protein for renal and hepatic tissues was determined using the method based on the Biuret reaction. Luck’s [47] method was used for catalase (CAT) activity. Superoxide dismutase activity was determined using the Misra and Fridovich method [48]. Glutathione-S-Transferase (GST) activity was as described by Habig et al. [49], while the reduced glutathione assay was according to Beutler et al. [50]. Lipid peroxidation was measured by measuring the formation of thiobarbituric acid reactive substance (TBARS), according to the method of Varshney and Kale [51]. The glutathione peroxidase (GPx) activity was determined by Rotruck et al. [52].

### 4.10. Histopathological Analysis

Histopathological analysis for renal and hepatic tissues using tissue sections stained with hematoxylin and eosin (H&E). Tissues from the experimental groups were fixed in 10% formaldehyde, dehydrated in graded alcohol, cleared in xylene, and finally embedded in paraffin before staining. The sections were viewed with a light Amscope microscope.

### 4.11. In Silico Studies

#### 4.11.1. Ligand Preparation

The three-dimensional structures of the eleven most abundant compounds present in the ethanolic extract of Leea guineensis, as revealed from the GC–MS analysis, were obtained from the PubChem (2020: http://pubchem.ncbi.nlm.nih.gov (accessed on 23 September 2023) database in SDF format. The compounds include squalene; Vitamin E; 1, 2, 3-benzentriol; pyrazole-5-carboxylic acid; 1,1,1,3,5,5,5-heptamethyl trisiloxane; beta sitosterol; beta tocopherol; thieno (2,3-C) furan-3-carbonitrile; benzo(h)quinoline; n-hexadecanoic acid; and curan 17-oic acid. Paracetamol was used as a reference control. The PyMol Molecular Graphics version 1.7.4 Schrodinger LLC, downloaded from pymol.org on 25th September, 2023, was used to convert the compounds in SDF format to PDB files. The ligands were then converted into pdbqt files, and the number of bonds that can be rotated was set to maximum using MGL Tools version 1.5.6 [53].

#### 4.11.2. Protein Preparation and Molecular Docking

The following proteins are known to play antioxidant roles under conditions of oxidative stress: Keap1 (Kelch-like ECH-associated-protein 1), Cul3 (Cullin 3), and Nrf2-mediated gene, of which the glutamate-cysteine ligase modifier subunit (GCLM) is an example. NAD(P)H: quinine oxidoreductase 1 (NQO1) and heme oxygenase 1 (HMOX1) were docked with the eleven major compounds present in the plant extract as revealed by GC–MS. The 3D structures of the mentioned target proteins were obtained from the RCSB Protein Data Bank (PDB) (www.rcsb.org) [54] in PDB format. The PDB ID codes of the respective proteins are shown in Table 5. The target protein structures were prepared using the Autodock Tools version 1.5.6 (the Scripps Research Institute, La Jolla, CA, USA) [53]. Water molecules, undesired protein chains, metal atoms, and the cocrystallized ligands were removed from the protein structure using Pymol, after which polar hydrogen atoms and Gasteiger charges were added, and docking grid maps were generated using the Autodock Tools v1.5.6 [53]. The compounds mentioned above were then docked at the active site of the proteins (depicted as a cleft when the protein was viewed in 3D, as revealed by Autodock MGL Tools 1.5.4). The grid box delineated the active site with coordinates and size, as shown in Table 5, considering the amino acids at the enzyme’s active site. The target was saved in the required file format (pdbqt). The docking calculations were conducted by command prompt using AutoDock Vina-1.1.2 [55], a modern-day docking software produced by the Molecular Graphics Laboratory.

PyMOL and Discovery Studio [56] were used for the docking analyses. BIOVIA Discovery Studio Visualizer version 19 (https://discover.3ds.com/discovery-studio-visualizer-download (accessed on 23 September 2023)) was used to view the binding intermolecular interaction of the compounds with the target proteins. The results are in (∆G, kcal/mol), i.e., the free energy value generated from binding the ligands to the respective proteins.

### 4.12. Statistical Analysis

Using one-way analysis of variance (ANOVA) followed by a post-hoc Tukey multiple comparison test (Graph Pad Prism software, Inc., San Diego, CA, USA), the data from the experiment were statistically evaluated and reported as mean ± standard deviation (SD); *p* < 0.05 values were deemed statistically significant.

## 5. Conclusions

This study shows that paracetamol-induced liver and kidney damage in rats is via oxidative stress. SLM, the standard drug used in the study, protected the organs from oxidative damage in paracetamol-induced toxicity, as evidenced by its ability to reverse some biochemical parameters altered by PCM. Similarly, LGE displayed protection against liver and kidney damage in PCM, bringing the levels of most biochemical parameters measured based on liver and kidney functions to those of healthy rats. Furthermore, treatment with *Leea guineensis* extract restored antioxidant parameters and significantly reduced MDA levels. Histopathological examination revealed that the extract also ameliorated paracetamol-induced liver and kidney damage, indicating both hepatoprotective and nephroprotective effects. It was theorized that the increased LGE concentration or extended extract administration period could lead to a complete reversal of hepatotoxicity and nephrotoxicity parameters, thus suggesting further studies. The limitations include short-term study duration, a lack of bioactive characterization, and the fact that specific molecular mechanisms such as Nrf2 and interleukin are not investigated. Further studies could explore different doses of *Leea guineensis* and varying treatment durations to assess whether a complete toxicity reversal can be achieved and maintained, as well as the isolation of active compounds and mechanistic and sex-based studies.

## Figures and Tables

**Figure 1 ijms-26-06142-f001:**
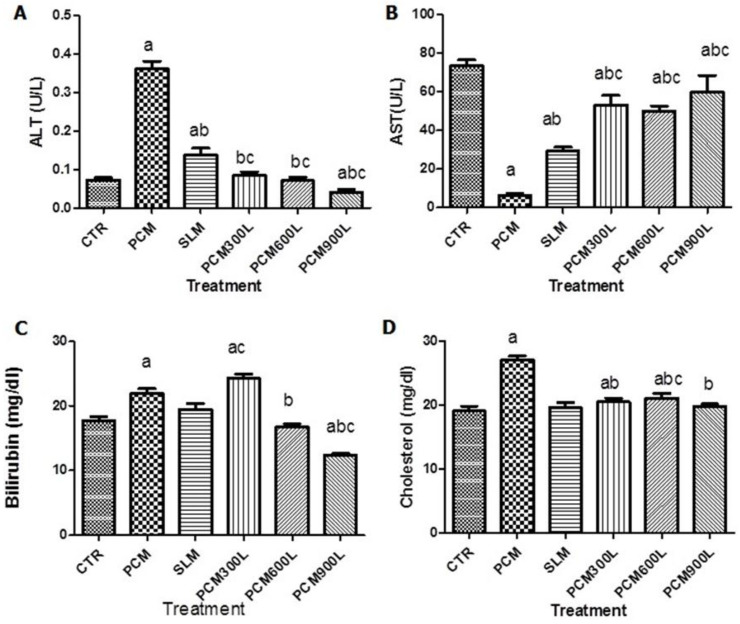
Effect of *Leea guineensis* extract on ALT (**A**), AST (**B**), bilirubin (**C**), and cholesterol (**D**) of rats’ plasma. CTR: Control; PCM: hepatotoxic; SLM: silymarin (standard drug); PCM300L: LGE extract 300 mg/kg; PCM600L: LGE extract 600 mg/kg; PCM900L: LGE extract 900 mg/kg. (a) Significant when compared to the control, (b) significant when compared to hepatotoxic control, and (c) significant when compared to silymarin. Each group comprises five rats.

**Figure 2 ijms-26-06142-f002:**
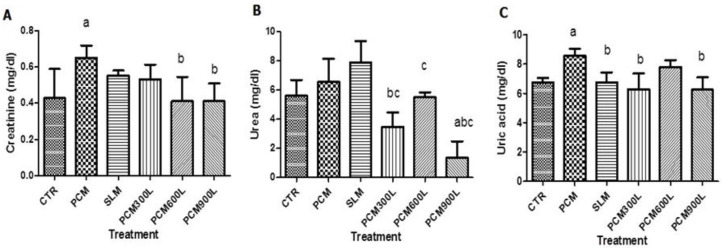
Effect of *Leea guineensis* ethanol extract on creatinine (**A**), urea (**B**), and uric acid(**C**) in the plasma of rats induced by paracetamol. CTR: healthy control; PCM: nephrotoxicity control; SLM: silymarin control; PCM300L: LGE extract 300 mg/kg; PCM600L: LGE extract 600 mg/kg; PCM900L: LGE extract 900 mg/kg. (a) Significant difference when compared to the control, (b) significant when compared to hepatotoxic control, and (c) significant when compared to silymarin. Each group comprises five rats.

**Figure 3 ijms-26-06142-f003:**
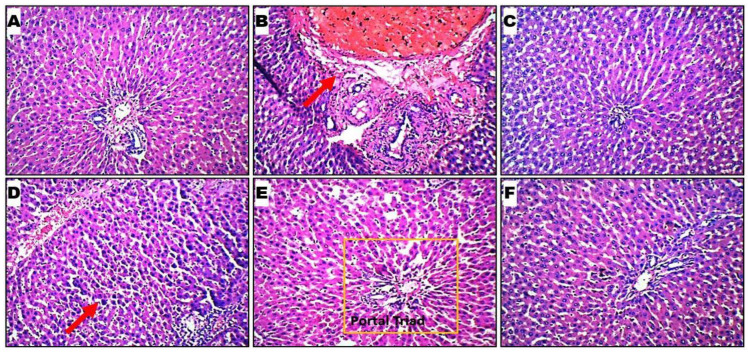
Hematoxylin and eosin staining at low magnification (×100) demonstrated panoramic views of a liver micromorphological section. Histological profiles of liver cells in 6 rat cohorts. (**A**) Healthy control; (**B**) hepatotoxic control showing massive inflammation of liver cells; (**C**) silymarin control at 200 mg/kg indicating inflammation of liver cells; (**D**) LGE at 300 mg/kg; (**E**) LGE at 600 mg/kg; (**F**) LGE at 900 mg/kg. Sub-Figures (**D**–**F**) demonstrate the healing of liver cells after the extract treatments toward paracetamol intoxication. The red arrow in (**B**) indicates the altered region of the liver histology, while in (**D**) it shows the healing of the liver architecture. The orange box indicates the portal triad.

**Figure 4 ijms-26-06142-f004:**
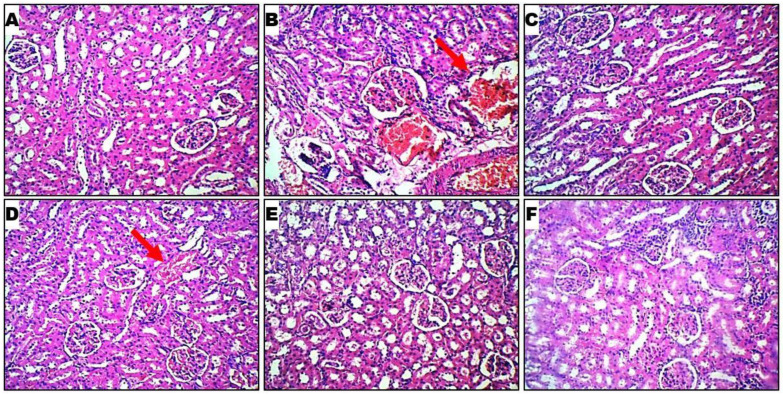
Panoramic views of the micromorphological kidney section as demonstrated by hematoxylin and eosin staining at low magnification (×100). Histological profiles of kidney cells in 6 rat cohorts: (**A**) healthy control; (**B**) hepatotoxic control showing massive inflammation of liver cells; (**C**) silymarin control at 200 mg/kg indicating inflammation of kidney cells; (**D**) LGE at 300 mg/kg; (**E**) LGE at 600 mg/kg; and (**F**) LGE at 900 mg/kg. Sub-Figures (**D**–**F**) demonstrate the healing of kidney cells after the extract treatments toward paracetamol intoxication. The red arrow shows the changes in the kidney morphology of the hepatotoxic control (**B**), while the red arrow in (**D**) shows that the kidney cells has been healed.

**Figure 5 ijms-26-06142-f005:**
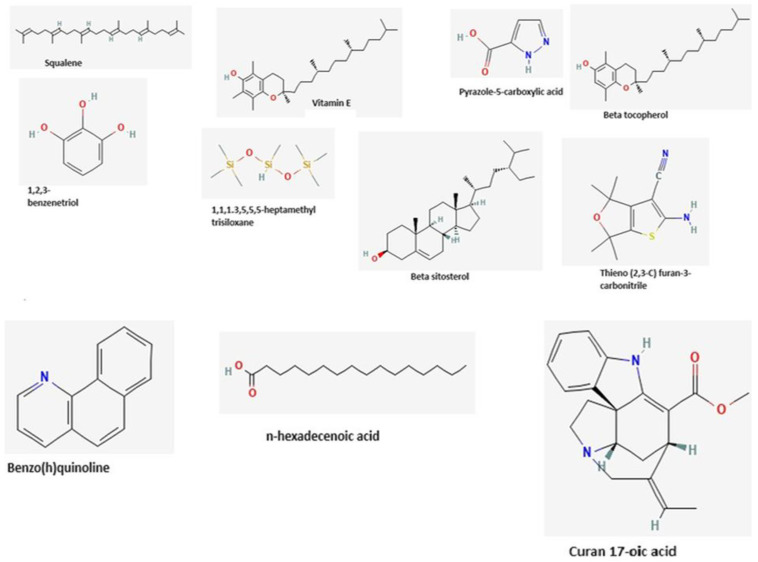
Structures of the major phytochemicals in the extract.

**Figure 6 ijms-26-06142-f006:**
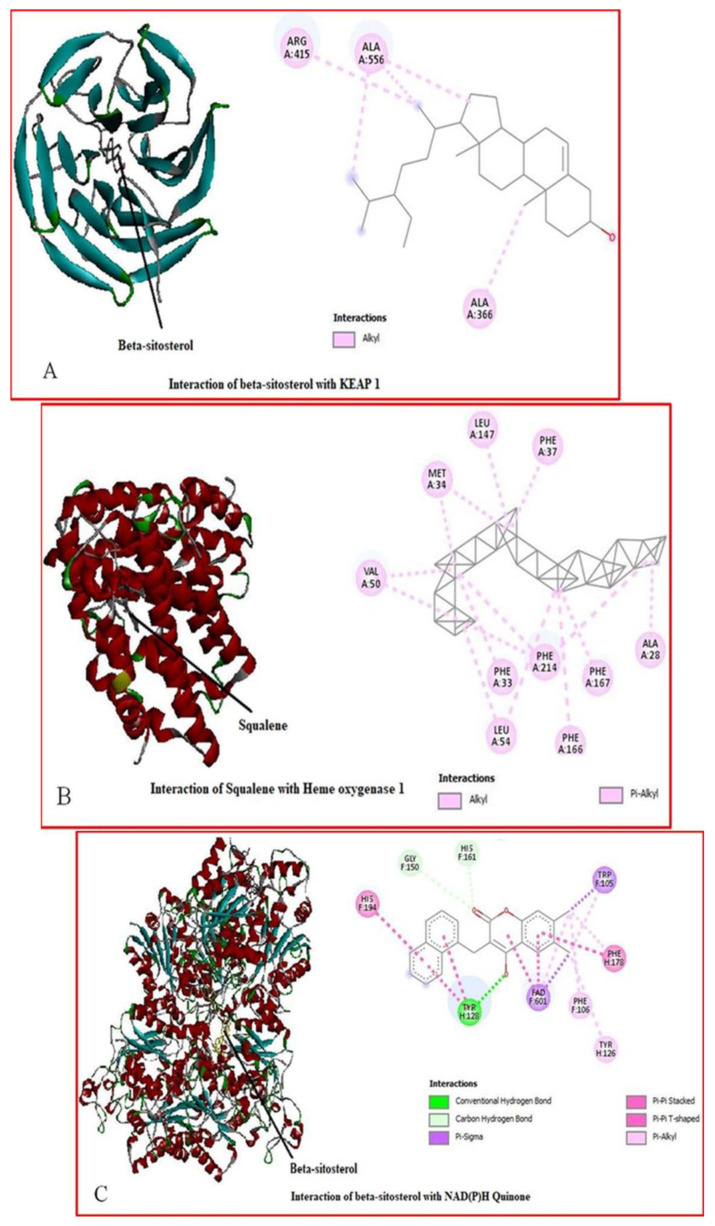
(**A**–**E**) Interaction networks. (**A**) Beta-sitosterol-bound KEAP 1, (**B**) squalene-bound. Heme Oxygenase 1, (**C**) beta-sitosterol-bound NAD(P)H Quinone, (**D**) beta-sitosterol-bound glutamate-cysteine ligase and (**E**) beta-sitosterol-bound Cullin 3. The protein structure is being depicted as a ribbon and the structure shown is that of the secondary structure in which barrels are shown in light blue colors and the helices present are depicted in red.

**Table 1 ijms-26-06142-t001:** Phytochemical analyses of the ethanol extract of *Leea guineensis* leaves.

Phytochemical Constituents	Test	Result
Alkaloids	Wagner	+
	Dragendorff	+
Tannins	FeCl_3_	++
Flavonoids	FeCl_3_	++
Saponins	Foam height	+
Phenolic	NH_4_ thiocyanate	+
Phlobatannins	HCl	+
Glycoside	Fehling’s solution	++
Carbohydrate	Molisch’s test	+
Protein	Biuret test	+

Qualitative detection of phytochemicals in *Leea guineensis* extract; (+) positivity, (++) strong positivity in the test.

**Table 2 ijms-26-06142-t002:** Summary of the body, liver, and kidney weights of animals.

Weight	Body (g)		Liver (g)		Kidney (g)			
TREATMENT	Initial	Final	Absolute	Relative		Absolute	Relative	
CTR	117.00 ± 7.91	193.40 ± 17.74	6.26 ± 0.84	0.033	±0.005	1.42 ± 0.08	0.007	±0.001
PCM	120.00 ± 10.85	195.80 ± 21.51	6.10 ± 1.29	0.031	±0.006	1.33 ± 0.17	0.007	±0.001
SLM	113.00 ± 11.34	187.00 ± 16.00	6.60 ± 0.54	0.035	±0.002	1.34 ± 0.17	0.006	±0.003
PCM + LGE (300 mg/kg)	114.40 ± 6.67	186.20 ± 10.96	5.83 ± 0.64	0.031	±0.004	1.16 ± 0.04	0.007	±0.002
PCM + LGE (600 mg/kg)	114.00 ± 8.86	185.60 ± 27.37	5.72 ± 0.97	0.035	±0.005	1.14 ± 0.06	0.007	±0.001
PCM + LGE (900 mg/kg)	106.60 ± 6.58	177.80 ± 17.41	5.93 ± 0.97	0.038	±0.006	1.36 ± 0.17	0.007	±0.001

Results are represented as Mean ± SD, n = 5.

**Table 3 ijms-26-06142-t003:** The effect of Leea guineensis extract on lipid, protein, and selected oxidative stress markers in PCM-impaired liver and kidney.

	TREATMENT	Total Protein	MDA	CAT	SOD	GPx	GSH	GST
**Liver**	CTR	4.18 (±0.44)	20.95 (±0.78)	1.61 (±0.00)	0.15 (±0.06)	1.80 (±0.29)	0.83 (±0.08)	0.09 (±0.04)
	PCM	1.50 (±0.24 ^a^)	22.63 (±2.44 ^a^)	4.48 (±0.01 ^a^)	0.05 (±0.03 ^a^)	1.50 (±0.38)	0.81 (±0.05)	0.14 (±0.04 ^a^)
	SLM	1.86 (±0.11 ^a^)	15.71 (±0.51 ^ab^)	3.62 (±0.00 ^ab^)	1.50 (±0.08 ^ab^)	1.37 (±0.33 ^b^)	0.84 (±0.08)	0.36 (±0.06 ^ab^)
	PCM + LGE (300 mg/kg)	0.30 (±0.02 ^ab^)	13.39 (±0.91 ^ab^)	5.25 (±0.04 ^ac^)	0.10 (±0.02 ^c^)	0.34 (±0.02 ^abc^)	0.46 (±0.02 ^abc^)	0.86 (±0.02 ^abc^)
	PCM + LGE (600 mg/kg)	1.60 (±0.46 ^a^)	14.79 (±3.92 ^ab^)	4.21 (±0.01 ^ab^)	0.22 (±0.04 ^b^)	1.64 (±0.22)	0.64 (±0.06 ^abc^)	0.19 (±0.01 ^ac^)
	PCM + LGE (900 mg/kg)	3.59 (±0.62 ^bc^)	17.87 (±1.59 ^ab^)	1.87 (±0.00 ^abc^)	0.10 (±0.02 ^b^)	1.14 (±0.04 ^a^)	0.80 (±0.08)	0.02 (±0.01 ^bc^)
**Kidney**	CTR	0.31 (±0.01)	2.33 (±0.05)	2.89 (±0.00)	0.30 (±0.03)	1.43 (±0.40)	0.79 (±0.09)	0.06 (±0.06)
	PCM	2.95 (±0.86 ^a^)	23.43 (±1.01 ^a^)	0.22 (±0.01 ^a^)	0.19 (±0.01 ^a^)	2.30 (±0.46)	0.69 (±0.11)	0.03 (±0.03)
	SLM	0.85 (±0.47 ^ab^)	14.62 (±0.4 ^ab^)	0.77 (±0.00 ^ab^)	0.23 (±0.09 ^ab^)	1.04 (±0.39 ^b^)	0.72 (±0.02)	0.08 (±0.06 ^ab^)
	PCM + LGE (300 mg/kg)	3.09 (±0.84 ^abc^)	25.81 (±1.42 ^ac^)	0.26 (±0.01 ^ac^)	0.10 (±0.01 ^abc^)	1.40 (±0.56)	0.57 (±0.05 ^a^)	0.02 (±0.01 ^abc^)
	PCM + LGE (600 mg/kg)	2.84 (±0.81 ^ac^)	12.09 (±1.60 ^abc^)	0.55 (±0.01 ^ab^)	0.11 (±0.01 ^abc^)	1.65 (±0.45 ^b^)	0.66 (±0.09)	0.05 (±0.05 ^bc^)
	PCM + LGE (900 mg/kg)	1.51 (±0.01 ^abc^)	5.03 (±0.04 ^abc^)	1.10 (±0.00 ^abc^)	0.14 (0.02 ^abc^)	1.25 (±0.27 ^b^)	0.74 (±0.09)	0.06 (±0.04 ^bc^)

Results are represented as Mean ± SD *n* = 5; a = significantly different (*p* < 0.05) compared to the normal group. b = significantly different (*p* < 0.05) when compared to PCM, c = significantly different (*p* < 0.05) when compared to SLM Group. Units: Total Protein (mg/g tissue), MDA (μmol MDA formed/mg protein), CAT (μmol/H_2_O_2_ consumed), SOD (Units/mg protein), GPx (μmol/mg protein), GSH (μmol/g tissue), GST (μmol-CDMB-GSH complex formed/mg protein).

**Table 4 ijms-26-06142-t004:** Heat map of the recorded docking scores (binding free energy kcal/mol) of the compounds in the ethanolic leaf extract of *Leea guineensis*.

		KEAP 1	Heme Oxygenase 1	NQO1	GCLM	Cullin-3
S/N	Ligand	5CGJ	6EHA	3JSX	3LVW	2MYM
1	Squalene	−6.1	−8.3	−7.8	−6.3	−4.8
2	Vitamin E	−6.9	−7.8	−9.1	−6.9	−6.1
3	1,2,3-benzenetriol	−5.6	−5.1	−5.7	−5.3	−3.8
4	Pyrazole-5-carboxylic acid	−6.4	−5.5	−5.8	−5.4	−4.4
5	1,1,1,3,5,5,5-heptamethyl trisiloxane	−5.9	−5.8	−5.8	−6.1	−3.7
6	Beta-sitosterol	−9.7	−7.8	−9.4	−7.8	−7.0
7	Beta tocopherol	−7.2	−6.3	−9.0	−6.5	−6.0
8	Thieno (2,3-C) furan-3carbonitrile	−7.3	−6.6	−7.1	−5.9	−4.5
9	Benzo(h)quinolinine	−6.8	−8.2	−8.1	−7.5	−5.5
10	n-hexadecanoic acid	−5.1	−6.2	−5.9	−5.1	−3.8
11	curan 17-oic acid	−8.5	−7.9	−9.1	−7.7	−5.6
12	Paracetamol	−5.9	−6.2	−6.0	−5.9	−4.3

**Table 5 ijms-26-06142-t005:** Molecular docking parameters and protein targets.

Protein	PDB ID	Grid Box Centre Coordinates	Grid Box Size
KEAP 1	5CGJ	center_x = 37.725 center_y = −11.616, center_z = 3.59	size_x = 28 size_y = 38, size_z = 38
Heme oxygenase 1	6EHA	center_x = 6.72 center_y = 6.155 center_z = 18.076	size_x = 66 size_y = 44 size_z = 92
NAD(P)H: QUINONE (NQ1)	3JSX	center_x = 20.644 center_y = −21.352 center_z = 27.171	size_x = 54 size_y = 74 size_z = 46
Glutamate-cysteine ligase	3LVW	center_x = 3.126 center_y = 36.637 center_z = −23.15	size_x = 54 size_y = 32 size_z = 40
Cullin 3	2MYM	center_x = 11.156 center_y = 9.862 center_z = 3.146	size_x = 36 size_y = 54 size_z = 50

## Data Availability

Data will be made available on request.

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
