# Peer review of "Hepatoprotective and Nephroprotective Effects of *Leea guineensis* Leaf Extract Against Paracetamol-Induced Toxicity: Combined Mouse Model-Integrated in Silico Evidence"

_ijms, 2025, doi:10.3390/ijms26136142_

Round 1
Reviewer 1 Report
Comments and Suggestions for Authors
This paper investigates the protective effects of leaf extract (LGE) against paracetamol (PCM)-induced liver and kidney damage in rats. The authors propose that PCM-induced toxicity is mediated by oxidative stress.
The study employs molecular docking to predict the binding affinities of LGE compounds to target molecules involved in PCM-induced hepatotoxicity and nephrotoxicity. This is complemented by an in vivo rat model where LGE or silymarin (SLM, a standard drug) were co-administered with PCM at various concentrations. The research evaluates hepatic and renal function, oxidative stress markers, and the safety of LGE. The findings suggest that LGE has the potential to ameliorate PCM's toxic effects.
A few minor revisions could further enhance the manuscript.
1.Table 3: Please ensure the number of rats for each experimental group is clearly indicated, consistent with the presentation in Table 4.
2. Figures 3 and 4: To improve data interpretation, kindly include the number of rats per group and the corresponding p-values directly on these figures.
3. Discussion of Molecular Docking Advantage: Given the established literature on paracetamol-induced nephrotoxicity and oxidative stress (e.g., "Paracetamol-induced nephrotoxicity and oxidative stress in rats: the protective role of Nigella sativa" and "Inhibition of Paracetamol-Induced Acute Kidney Damage in Rats Using a Combination of Resveratrol and Quercetin"), it would be highly beneficial to include a discussion highlighting the specific advantages of employing molecular docking in this study, particularly from a clinical application perspective, compared to solely relying on in vivo methods."
Reviewer 2 Report
Comments and Suggestions for Authors
Comments
The manuscript with entitled “ Hepatoprotective and nephroprotective effects of Leea guineensis leaves extract against Paracetamol-induced toxicity: A com-bined mouse model-integrated in-silico evidence”.
- This manuscript studied the epatoprotective and nephroprotective effects of Leea guineensis leaves extract against Paracetamol-induced toxicity by experimental validation. It is interesting.
- In all manuscript, the Latin name format of the species should meet the requirements.
- In the methodology, all chemicals, reagents and kits should be in the list and provide source, concentration, the cargo number of the reagent.
- Ethical license and license number should be provided.
- Paracetamol is clinically available? what is the dose used? What is the relationship between the dose of clinically and the dose used in this study? What are the characteristic signs of the model of Paracetamol-induced toxicity? How does the author determine the success of the model of Paracetamol-induced toxicity? The author should have clear indicators of successful model establishment. Recommendation: The author to provide supporting materials and additional clarification in the article.
- What are the extraction rates of Leea guineensis leaves?What is base of the dose of Leea guineensis leaves extract? Recommendation: The author to provide additional clarification.
- In Figure 2. Paracetamol increased ALT activity compared to the control group, However, Paracetamol decreased AST activity compared to the control group, Why? The author to provide additional clarification.
- In Figure 3B. Paracetamol goup is not significant difference when compared to control?
- In molecular docking analyses, how does the author evaluate that molecular docking has reached the best level?
- In the conclusions, what are the specific limitations of the article and future research directions? The author should focus on discussing?
- There a multiple grammar and style errors, wrongly constructed phrases and scientifically incorrect expressions. Such as, Kaep1, (ubiquitin-proteasome system) [56]……
- The format of references is not consistent, and it does not meet the requirements of the journal.
There a multiple grammar and style errors, wrongly constructed phrases and scientifically incorrect expressions. Such as, Kaep1, (ubiquitin-proteasome system) [56]……
Round 2
Reviewer 2 Report
Comments and Suggestions for Authors
None